# Variable Molecular Weight Polymer Nanoparticles for Detection and Hyperthermia-Induced Chemotherapy of Colorectal Cancer

**DOI:** 10.3390/cancers13174472

**Published:** 2021-09-05

**Authors:** Santu Sarkar, Nicole Levi

**Affiliations:** Department of Plastic and Reconstructive Surgery, Wake Forest University School of Medicine, Winston-Salem, NC 27157, USA; santchem83@gmail.com

**Keywords:** hyperthermia, nanoparticles, chemotherapy, colorectal cancer, fluorescent detection

## Abstract

**Simple Summary:**

The purpose of this work was to evaluate the development of polymer-based nanoparticles that can both generate heat and be used for fluorescence detection. The nanoparticles were used against luminescent colorectal cancer cells that were either sensitive or resistant to the chemotherapy drug, oxaliplatin. The fluorescence of the nanoparticles indicates that they are internalized within the cells for heat generation. Mild heating makes oxaliplatin-resistant cancer cells responsive to chemotherapy, and the nanoparticle-induced hyperthermia causes cell death in a few minutes, compared to classical bulk heating, which takes a few hours. Changes in the luminescence of the cancer cells can be used to determine the thermal dose induced by the nanoparticles, which may be correlated with the cell viability and therapeutic response.

**Abstract:**

Oxaliplatin plays a significant role as a chemotherapeutic agent for the treatment of colorectal cancer (CRC); however, oxaliplatin-resistant phenotypes make further treatment challenging. Here, we have demonstrated that rapid (60 s) hyperthermia (42 °C), generated by the near-infrared stimulation of variable molecular weight nanoparticles (VMWNPs), increases the effectiveness of oxaliplatin in the oxaliplatin-resistant CRC cells. VMWNP-induced hyperthermia resulted in a higher cell death in comparison to cells exposed to chemotherapy at 42 °C for 2 h. Fluorescence from VMWNPs was observed inside cells, which allows for the detection of CRC. The work further demonstrates that the intracellular thermal dose can be determined using cell luminescence and correlated with the cell viability and response to VMWNP-induced chemotherapy. Mild heating makes oxaliplatin-resistant cancer cells responsive to chemotherapy, and the VMWNPs-induced hyperthermia can induce cell death in a few minutes, compared to classical bulk heating. The results presented here lay the foundation for photothermal polymer nanoparticles to be used for cell ablation and augmenting chemotherapy in drug-resistant colorectal cancer cells.

## 1. Introduction

Colorectal cancer (CRC) is the fourth leading cause of cancer-related deaths, with few available treatment options [1,2,3]. Recent clinical studies showed that hyperthermia (39–42 °C) is an effective adjuvant therapeutic technique along with radiotherapy and/or chemotherapy, and that specifically 42 °C is routinely used clinically with intraperitoneal chemotherapy for the treatment of CRC [4,5,6,7,8,9]. Among the available chemotherapeutic drugs, oxaliplatin is a cornerstone for the treatment of CRC. One of the major disadvantages is that, due to the intermittent exposures of oxaliplatin, the cells become chemo-resistant, complicating treatment [10,11,12]. Hyperthermia increases the drug uptake by affecting cell membranes and producing drug-induced DNA damage, leading to enhanced tumor cell death [13,14,15,16]. It has been demonstrated that hyperthermia is synergistic with oxaliplatin for treating CRC [9,17,18]. In order to improve the precision of the technique, instead of using a bulk carrier fluid along with a heat exchanger in the traditional hyperthermia delivery, photothermal nanoparticles can be used to deliver more specific and effective hyperthermia.

Photothermal nanoparticles (NPs) that absorb light and generate heat have been extensively studied for both cancer cell ablation and mild hyperthermia. The most common photothermal agents are metallic; however, recent developments in the field of semiconducting polymers have instigated their evaluation as photothermal nanoparticles [19,20,21,22,23,24,25,26,27]. Recently, our team developed variable molecular weight nanoparticles (VMWNPs) produced from the oligomer and high MW segments of a single polymer, poly[4,4-bis(2-ethylhexyl)-cyclopenta[2,1-b;3,4-b′]-dithiophene-2,6-diyl-alt-2,1,3-benzoselenadiazole-4,7-diyl] (PCPDTBSe), and have demonstrated that they are a promising photothermal agent for the ablation of breast cancer [28]. VMWNPs generated heat upon 800 nm laser irradiation and produced fluorescence emission at 825 nm upon excitation with 550 nm. As shown in Scheme 1a, the facile synthesis of VMWNPs occurs using a nanoprecipitation method. The high molecular weight (HMW) fraction is capable of heat generation through electron–hole recombination, whereas the oligomer fraction is capable of fluorescence emission, as shown in Scheme 1b. Therefore, VMWNPs can be used to detect CRC through fluorescence imaging, as well as provide hyperthermia delivery. Here, we have explored the synergistic effect of oxaliplatin and hyperthermia generated by the photothermal VMWNPs for augmenting oxaliplatin in sensitive and resistant CRC cells.

## 2. Materials and Methods

### 2.1. Cells and Materials

CT-26 WT-Fluc-Neo, mouse CRC cells were purchased from Imanis Life Sciences. HT-29 and RKO human CRC cells were purchased from American Type Culture Collection. We previously developed oxaliplatin-resistant (OxR) cells for comparison to parental (oxaliplatin-sensitive (OxS) cells, as explained by McCarthy, et al. [29]. Dulbecco’s Modified Eagle Medium (DMEM) supplemented with penicillin and streptomycin, L-glutamine, 400 µg/mL G418 and with or without 10% Fetal Bovine Serum (FBS) was obtained from Gibco. McCoy’s Medium with penicillin and streptomycin, L-glutamine and with and without 10% FBS, was purchased from Gibco. Oxaliplatin, 1,3-diphenylisobenzofuran (DPBF) and Pluronic F-127 were purchased from Sigma-Aldrich. 4′,6-diamidino-2-phenylindole (DAPI) and Alexa Fluor^®^ 488 were purchased from Abcam. Tetrahydrofuran (THF) was purchased from Acros Organics. CellTiter 96^®^ AQueous One Solution Cell Proliferation Assay (MTS) was obtained from Promega. Phosphate-buffered saline (PBS; 1×) buffer with pH 7.4 was prepared and sterilized before use. 4,7-Dibromo-2,1,3-benzoselenadiazole and 4,4-Bis(2-ethyl-hexyl)-4H-cyclopenta[2,1-b:3,4-b′]dithiophene were obtained from TCI America.

PCPDTBSe was synthesized following published procedures [30]. Briefly, 4,4-Bis(2-ethylhexyl)-2,6-bis(trimethylstannyl)-4H-cyclopenta [2,1-b;3,4-b′]-dithiophene (1.5 mmol) and 4,7-dibromo-2,1,3-benzoselenadiazole (1 mmol) were combined in anhydrous toluene followed by the addition of Pd (PPh_3_)_4_ (5 mol%) and stirred at 110 °C for 24 h to obtain a mixture of oligomer, low molecular weight (MW) and high MW polymer fractions. The polymer fractions were separated by Soxhlet extraction using methanol (3 h), hexane (6 h) and chloroform (6 h). The methanol, hexane and chloroform fractions were evaporated to collect oligomer, low MW and high MW polymer fractions, respectively.

Absorbance spectra of the oligomer and high MW PCPDTBSe in THF were recorded using a Mettler Toledo UV-Vis spectrophotometer. Fluorescence spectra of the polymer fractions were obtained using a TECAN M200 Infinite plate reader with λ_ex_ = 550 nm.

### 2.2. Preparation and Characterization of VMWNPs

VMWNPs were prepared by nanoprecipitation following the methods described by Sarkar et al. [28]. Briefly, 1 mg of the high MW (HMW) and 2 mg of the oligomer PCPDTBSe were mixed in 2 mL THF and added to an 8 mL aqueous solution of Pluronic F-127 under horn sonication (20% amplitude, 110 s). Similar techniques were used to develop nanoparticles composed of only the oligomer or only the HMW fractions. THF was evaporated and the particles were sterilized by autoclaving prior to being centrifuged at 7500 rpm for 30 min to pellet large NPs. The supernatant was centrifuged at 14,000 rpm for 10 h to collect small nanoparticles, which were used for the experimental studies.

### 2.3. Heating Methods

A 300 μL solution of VMWNPs in cell culture media was irradiated with an 800 nm laser (Cube TM continuous-wave diode laser from Summus Medical Laser, Inc., Franklin, TN, USA (1 or 3 W, beam diameter—1 cm)). A Fluke 714 thermometer and a type k 80Pk-1 bead probe wire thermocouple were used to measure the temperature of the solutions immediately before and after laser application. It was found that 1 W laser stimulation of VMWNPs for 60 s generated 42 °C for hyperthermia, and these parameters were used to augment chemotherapy. Photothermal ablation (T > 45 °C) was accomplished using either longer time with 1 W, or else 3 W of laser power. A fiber optic thermocouple (Qualitrol Neoptix^®^ and Nomad thermometer, Fairport, NY, USA) was alternatively used to measure continuous temperature increases in VMWNPs solutions over time.

### 2.4. VMWNPs Cytotoxicity

To evaluate VMWNPs’ cytotoxicity to the OxS and OxR CT-26, HT-29 and RKO cells, they were individually plated at 5000 cells/well in 96 well plates and cultured for 24 h. Two hundred microliters of VMWNPs solution with varying concentrations (0, 25, 50, 100, 250 and 500 μg/mL) were added to triplicate wells and incubated for 24 h at 37 °C. Cells with no treatment were used as controls. Nanoparticle solutions were removed after 24 h, cells were washed with PBS and viability was quantified using Promega’s Cell Titer 96 AQ_ueous_ assay kit.

### 2.5. Singlet Oxygen Measurements

DPBF was used to quantify generation of singlet oxygen upon laser irradiation. A 1 mg/mL DPBF solution in methanol was prepared and stored in dark. Absorbance (418 nm) of a 65 µg/mL DPBF solution in water was checked before and after three consecutive laser irradiation stimulation periods (800 nm, 1 W, 60 s) using a TECAN M200 Infinite plate reader before and after three consecutive laser treatments. Absorbance of DPBF solutions was also measured for samples without laser stimulation. Similar experiments were conducted with solutions containing 25 µg/mL VMWNPs, oligomeric or HMW nanoparticles and 65 µg/mL DPBF.

### 2.6. Intracellular Concentration of VMWNPs

To quantify intracellular concentration of VMWNPs, first a calibration curve was developed. The absorbance of 300 μL of 0, 25, 50, 100, 250 and 500 μg/mL VMWNPs in media with and without serum was measured at 760 nm. Then, OxS and OxR cells were plated at 20,000 cells/well in FBS containing media in a 48-well plate and cultured for 24 h. The media was aspirated and 300 μL of varying concentrations (0, 25, 50, 100, 250 and 500 μg/mL) of VMWNPs in FBS-containing media were added in triplicate. After 24 h of incubation, the nanoparticle solutions were removed and estimated by recording their absorbance at 760 nm. Intracellular nanoparticle uptake was quantified by subtracting the NP concentration of the supernatant from the total concentration of NPs added. The same technique was also applied to cells that had been previously starved of serum by culturing in FBS-free media.

### 2.7. Visualization of VMWNPs

OxS and OxR CT-26 CRC cells were plated onto collagen-coated cover slips and incubated for 24 h with 0 or 100 µg/mL of VMWNPs in either FBS-free media. Following incubation, VMWNPs solutions were removed and cells were washed twice with cold PBS, fixed with 4% paraformaldehyde and then stained with Alexa-fluor-488 and 4′,6-diamidino-2-phenylimdole (DAPI). An Olympus FV1200 SPECTRAL Laser scanning Confocal Microscope (Olympus IX83 inverted platform) was used to collect images of the cells and VMWNPs. To visualize the fluorescence from VMWNPs in vivo, a freshly euthanized female Balb/C mouse was intraperitoneally injected with 100 µL or a 250 µg/mL VMWNPs solution and imaged using a Perkin Elmer Caliper in vivo imaging system (IVIS). Lamp excitation of 465 nm was used and the indocyanine green (ICG) filter was used to capture emitted light above 695 nm.

### 2.8. In Vitro Photothermal Effect of VMWNPs

OxS and OxR CT-26 cells were used to evaluate the effect of intracellular VMWNPs on inducing cell death by photothermal ablation. The cells were plated at 10,000 cells/well in a 48-well plate in FBS-free media and cultured 24 h. They were then incubated with 250 µg/mL of VMWNPs in FBS-free media for 24 h, following washing to remove non-internalized VMWNPs, and exposed to 3 W of 800 nm light for 120, 180, 240 or 300 s. Following photothermal treatment, cells were washed and incubated for 24 h before cell viability was quantified using Promega’s Cell Titer 96 AQ_ueous_ assay kit.

To evaluate the effect of extracellular VMWNP-induced photothermal ablation, OxS and OxR CT-26, HT-29 and RKO CRC cells were seeded at 20,000 cells/well in a 48-well plate and cultured for 24 h. VMWNPs solutions (0, 25, 50 and 100 μg/mL) were added to triplicate wells and incubated for 20 min at 37 °C immediately prior to laser exposure. Plates were maintained at 37 °C during laser exposure by placing them on a heat block, during which they were stimulated with 800 nm laser (1 W) for 60 s and 120 s, respectively. Cells were incubated for 24 h and viability was quantified using Promega’s Cell Titer 96 AQ_ueous_ assay kit.

### 2.9. Luminescent Monitoring of Thermal Dose Following VMWNPs-Induced Photothermal Ablation

Thermal dose was determined by loss of intracellular luminescence following photothermal treatment. OxS and OxR CT-26 WT-Fluc-Neo CRC cells, which are luminescent, were seeded at 20,000 cells/well in 48-well plates and cultured for 24 h. VMWNPs solutions (0, 25, 50 and 100 μg/mL) were added to the cells in triplicate and incubated for 20 min at 37 °C immediately prior to laser exposure. Cells were then stimulated with 800 nm laser (1 W) for 60 s and 120 s, during which they were kept on a heat block set at 37 °C. Immediately after laser exposures, the treatment solutions were removed and cells were washed with PBS. Three hundred microliters of 150 μg/mL luciferase solution in DMEM media was added to each well and the plates were incubated at 37 °C for 10 min. Luminescence was measured using a FilterMax F5 Multi-Mode Microplate Reader (1000 ms integration time), and luminescence intensities were normalized to the relative luminescence at 37 °C. Percentage loss of luminescence was calculated for each concentration of VMWNPs. Thermal dose was correlated to the loss of luminescence using $CEM43=\int_{0}^{t}R^{\left(43-T\right)}\;dt\;\left(\min\right)$ to calculate cumulative equivalent minutes at 43 °C (CEM43). Here, t is time in minutes, R is a correction for the number of minutes required to achieve an isoeffect for each degree step away from 43 °C (R = 0.25 when T < 43 °C and R  =  0.5 when T > 43 °C) and T is the temperature [14,29,31,32]. Calculated CEM43 at respective temperatures was then compared to CEM43 determined using the luminescence changes following laser stimulation of cells with VMWNPs.

### 2.10. Photothermal Chemotherapy

OxS and OxR cell lines were plated at 20,000 cells/well in 48-well plates and cultured for 24 h. To evaluate the benefits of hyperthermia on oxaliplatin effectiveness, triplicates of cells were treated with concentrations of oxaliplatin (5, 25, 100 and 300 μM) in two different plates held at 37 °C or 42 °C for two hours. After oxaliplatin exposure, cells were washed with PBS, incubated for 48 h and cell viability was quantified using Promega’s Cell Titer 96 AQ_ueous_ assay kit.

Oxaliplatin efficiency against OxS and OxR cells was also assessed in the presence of VMWNPs, laser and NPs + laser (to generate mild hyperthermia at 42 °C). Cells were seeded at 20,000 cells/well in 48-well plates and treated with 0, 5, 25, 100 and 300 μM oxaliplatin in triplicate, in the independent presence of 25 µg/mL VMWNPs, laser (800 nm, 1 W, 60 s) or NPs + laser. During the two hours of oxaliplatin exposure, three laser treatments were applied for each of the ‘laser’ and ‘NPs + laser’ treatment groups for a total of three 60 s intervals, with a 20 min normothermic recovery time. The rationale for applying three laser applications was to use the rapid heating from the NPs to aid in cell membrane permeabilization for oxaliplatin transport into cells and then return to normothermia for drug retention. Plates were maintained at 37 °C during laser exposure by placing them on a heat block. After two hours, treatment solutions were aspirated and cells were washed with PBS, followed by addition of 300 μL of media to each well. After 48 h of incubation, cell viability was quantified using Promega’s Cell Titer 96 AQ_ueous_ assay kit.

## 3. Results

### 3.1. Preparation and Characterization of PCPDTBSe NPs and VMWNPs

PCPDTBSe was synthesized through Stille coupling following reported literature procedures and characterized (Figure 1a) [33]. The different molecular weight fractions of PCPDTBSe showed that the oligomer NPs had a distinct absorption at 550 nm, whereas the high MW fraction had an absorption maximum at 760 nm (Figure 1b). The oligomeric NPs showed a near-infrared fluorescence, with the peak maxima at 750 nm (λ_ex_ = 550 nm), but the high MW fraction had a minimal fluorescence (Figure 1c). VMWNPs were prepared from the oligomer and high MW fraction using a nanoprecipitation method (Scheme 1a). The oligomer fluorescence emission was quenched due to the spectral overlap of the absorbance spectrum of the high MW PCPDTBSe and emission spectrum of the oligomer (Figure 1c). Quantum yield (QY) is a useful measurement to gauge the fluorescence of nanoparticles, and we have previously found the QY of oligomeric NPs to be 0.27 and VMWNPs to be 0.077, which, although low in value, allows for fluorescence detection in vitro [28]. The average hydrodynamic diameter of VMWNPs was determined to be 80 nm (Figure 1d). The zeta potential of the nanoparticles was −5.27 mV (Appendix A), and the particles were stable in water and serum-containing media. VMWNPs depicted an increase in temperature with increasing concentrations of VMWNPs (Figure 1e). A 42 °C elevation in temperature was observed by the laser irradiation of 100 µg/mL nanoparticles. To generate photothermal hyperthermia, varying concentrations of VMWNPs were stimulated with an 800 nm, 1 W laser for 60 and 120 s independently. Upon 60 s of a 1 W laser exposure to 0, 25, 50 and 100 µg/mL NPs, temperature increments of 6.55, 10.25 and 14.85 °C were obtained, respectively (Appendix A). With a 1 W laser at 120 s, higher temperatures of 12.45, 16.95 and 22.6 °C were found (Appendix A). Continuous measurements of temperature increases were recorded with 25, 50 and 100 µg/mL NPs upon 800 nm (1 W) laser irradiation for 60 s, further demonstrating that increasing concentrations produce increased temperatures of VMWNPs solutions (Appendix A).

### 3.2. Augmented Response to Oxaliplatin Using Hyperthermia

Resistance against oxaliplatin was developed by the continuous exposure of oxaliplatin to the parental (OxS) CT-26, HT-29 or RKO cells [29]. To assess how hyperthermia enhanced oxaliplatin efficiency, cells were incubated with oxaliplatin at either 37 °C or 42 °C for two hours. Cell death increased with increasing concentrations of oxaliplatin at both temperatures, and a greater loss in viability was noticed at 42 °C than at 37 °C, most likely due to a higher drug uptake at the elevated temperature. OxS CT-26 cells displayed an insignificant decrease in viability up to 50 μM oxaliplatin at 37 °C and 42 °C. At both 37 and 42 °C, there was a statistically significant reduction in the OxS population at 100 and 300 µM (Figure 2a), although 42 °C provided a statistical advantage only at 100 µM. In contrast, OxR CT-26 cells at 42 °C provided a statistically significant reduction in viability compared to 37 °C for all groups except 100 µM oxaliplatin. The onset of oxaliplatin-induced cell death was not evident until 100 µM, and a 51% and 25% reduction in viability was observed at 300 μM at 37 °C and 42 °C, respectively (Figure 2b).

OxS RKO cells were reduced at 42 °C compared to 37 °C (Figure 2c). In the presence of 25, 100 and 300 μM oxaliplatin, decreases in cell viability of 50%, 68% and 99% were found at 37 °C (Figure 2c). At 42 °C, 63%, 82%, 93% and 97%, reductions were observed at 5, 25, 100 and 300 μM oxaliplatin, respectively (Figure 2c). OxR RKO cells had an increased viability for 5, 25 and 100 µM at 37 °C, and only had a decrease at 100 µM with 42 °C. Only 300 μM oxaliplatin resulted in 67% and 25% reductions in cell viability at 37 °C and 42 °C, respectively (Figure 2d). The increase in cell viability for OxR RKO cells at lower oxaliplatin concentrations was an effect that was observed multiple times with this oxaliplatin-resistant cell line. The phenomenon may be described as a hormetic effect, wherein low concentrations of a toxic agent can stimulate cell growth [34,35,36]. It is an interesting observation that OxR HT-29 cells did not exhibit this effect. This result further supports the mechanisms by which chemotherapy resistance exacerbates CRC recurrence and progression.

OxS HT-29 cells had a decreased viability with an increasing oxaliplatin concentration. Beginning at 5 µM, there were 25, 42, 53 and 88% reductions with oxaliplatin at 5, 25, 100 and 300 µM delivered for 2 h at 37 °C. There were further reductions (31, 58, 86 and 82%) when oxaliplatin was provided at 42 °C, although mild hyperthermia only provided a statistical advantage at 25 and 100 µM oxaliplatin (Figure 2e). These results are in contrast to OxR HT-29 cells, which have a limited reduction in cell viability at 5 and 25 µM oxaliplatin (Figure 2f). There was a 37 and 52% reduction in viability for cells treated with 100 or 300 µM oxaliplatin at 37 °C. Only 42 °C and 300 µM demonstrated a statistically significant difference compared to OxR HT-29 cells treated at 37 °C, resulting in a 72% reduction. Mild hyperthermia (42 °C) confers an advantage at lower oxaliplatin concentrations in OxS RKO cells, whereas this effect is only observed at higher concentrations in OxR HT-29 cells. The mouse CT-26 cells and both human RKO and HT-29 cells that were oxaliplatin-resistant demonstrated resistance compared to OxS cells.

### 3.3. VMWNPs Cytotoxicity

The cytotoxicity of VMWNPs against OxS and OxR CT-26, HT-29 or RKO CRC cells for 24 h is shown in Appendix A. VMWNPs produced no considerable cytotoxicity towards OxS and OxR CT-26 cell lines up to a concentration of 100 μg/mL (Appendix A). A 39% and 52% reduction in cell viability was observed with OxS CT-26 cells at 250 μg/mL and 500 μg/mL. A similar phenomenon was observed in OxR CT-26 cells with reductions of 34% and 50% with 250 μg/mL and 500 μg/mL of the VMWNPs. OxS RKO cells had a 35% reduction with 25 and 50 μg/mL of VMWNPs, whereas an 81% decrease was found with 100 µg/mL NPs (Appendix A). No viable cells were observed at higher concentrations. OxR RKO cells did not show a reduction in cell viability with 25 µg/mL VMWNPs, but showed 26% and 76% decreases with 50 and 100 µg/mL NPs (Appendix A). Similar to the OxS RKO cells, no viable cells were detected at higher concentrations. The HT-29 cell response to VMWNPs was more closely aligned with that of CT-26 cells. There was only a significant reduction in viability at 250 µg/mL, resulting in 67 and 51% reductions for OxS and OxR HT-29 cells, respectively (Appendix A).

### 3.4. Quantification of Singlet Oxygen Generation by VMWNPs

The generation of singlet oxygen by VMWNPs was monitored using DPBF as a singlet-oxygen-specific trap. DPBF has a 90 absorption at 418 nm and, upon reaction with singlet oxygen, loses its absorbance intensity. Therefore, a reduced absorbance of DPBF indicates the generation of singlet oxygen species [37]. No change in absorbance intensity was observed with water or without the laser (Appendix A). To evaluate the singlet oxygen generation from VMWNPs, the absorbance intensity of a mixture of DPBF and VMWNPs solutions was monitored in the presence and absence of laser stimulation. As shown in Appendix A, the absorption intensity of the solution containing DPBF and VMWNPs was 0.77, but after three consecutive laser irradiations, the absorption intensity was reduced to 0.24, whereas in the absence of laser stimulation, the absorbance decreased to 0.61. Nanoparticles composed of oligomeric PCPDTBSe had a reduction of 72% with laser stimulation, compared to only a 34% reduction without. VMWNPs had a 64% reduction with laser stimulation, compared to 21% reduction without. This indicates that reactive oxygen species stem from the oligomeric fraction of PCPDTBSe. In the absence of laser stimulation, HMWNPs had an absorbance decrease of only 9%, compared to VMWNPs (21%) and oligomeric NPs (34%). The generation of reactive oxygen species might be the reason for cytotoxicity of the VMWNPs, as shown in Appendix A.

### 3.5. Quantification of Intracellular VMWNPs

The cellular uptake of VMWNPs was quantified with 24 h of incubation of 25–500 µg/mL, delivered in media with or without FBS (without FBS was used to promote cellular uptake by serum starving the cells). Previous literature has demonstrated that serum starving cells, or failing to provide FBS, can further facilitate the uptake of nanoparticles [36,38]. With FBS in the media, OxS CT-26 cells had 0 µg/mL when exposed to 25 or 50 µg/mL of VMWNPs (Appendix A). They had 3, 13 and 17 µg/mL when exposed to 100, 250 or 500 µg/mL of VMWNPs. After OxS CT-26 cells were serum starved, the approximate intracellular concentrations were determined as 4.7, 7.7, 12.7, 25.9 and 46.1 μg/mL upon incubation with 25, 50, 100, 250 and 500 μg/mL of VMWNPs, respectively. OxR CT-26 cells (Appendix A) have intracellular NPs of 3.8, 2.2, 4.3, 17.7 and 3.8 μg/mL with FBS-containing media, and 1.8, 4.2, 6.5, 13.7 and 18.9 μg/mL for FBS-free media, upon incubation with 25, 50, 100, 250 and 500 μg/mL of VMWNPs, respectively. The absence of FBS appears to drive VMWNP internalization.

As shown in Appendix A, intracellular concentrations for OxS RKO cells treated with 0 up to 100 µg/mL resulted in no measurable intracellular concentration when delivered in FBS-containing media, and 11.6 and 61.3 µg/mL for cells treated with 250 and 500 µg/mL. OxS RKO cells treated with VMWNPs in serum-free media had 2.4, 4.2, 6.1, 18.2 and 25.6 µg/mL of intracellular NPs. OxR RKO cells had 0µg/mL of intracellular VMWNPs regardless of the addition or absence of FBS for 25, 50 and 100 µg/mL (Appendix A). At a dose of 250 µg/mL, only serum-starved cells had intracellular VMWNPs, at a concentration of 12.3 µg/mL. VMWNPs in FBS-containing media at a concentration of 500 µg/mL had 11.1 µg/mL of intracellular VMWNPs compared to serum-starved cells, which had 32.7 µg/mL.

### 3.6. Imaging of VMNPs

OxS and OxR CT-26 cells were serum starved before they were incubated with 100 µg/mL VMWNPs. The in vitro uptake of VMWNPs was visualized through confocal microscopy. Figure 3 shows that red VMWNPs were internalized and visible around the nucleus in OxS and OxR CT-26 cells. Cells without VMWNPs do not exhibit red fluorescence due to the absence of VMWNPs. Interestingly, the OxR CT-26 cells seem to have increased internal VMWNPs, although this observation conflicts with the results of Appendix A, which determined that OxS CT-26 cells had a higher intracellular concentration than OxR CT-26 cells. The fluorescence of VMWNPs is also visible in vivo, as demonstrated in Appendix A. This figure shows a mouse with no tumor burden and an intraperitoneal delivery of a 100 µL volume of 250 µg/mL VMWNPs.

### 3.7. Photothermal Response of Intracellular VMWNPs

It was estimated that the incubation of 250 μg/mL VMWNPs in CT-26 OxS and OxR might provide sufficient intracellular nanoparticles for photothermal hyperthermia without producing significant cytotoxic effects. From Appendix A, this dose provides 25 and 13.7 µg/mL of VMWNPs in OxS and OxR cells, respectively. Therefore, OxS and OxR CT-26 CRC cells were incubated with 250 μg/mL VMWNPs in serum-free media for 24 h. After removing excess nanoparticles, cells were stimulated with an 800 nm 3 W laser for varying times of exposure in order to induce cell ablation. From Figure 1e, 15 µg/mL induced 14 °C at 60 s of 3 W exposure, which is above the ΔT = 13 °C needed for ablation. To help ensure ablation for longer times of laser stimulation, up to 300 s were used. OxS cells had a reduced viability with VMWNPs and an increasing laser exposure, whereas the laser alone produced no cell death, and actually resulted in a cell stimulation effect, as shown in Appendix A. The increase in cell viability at 240 s and 300 s of laser stimulation with internalized VMWNPs may indicate that these cells did not internalize a sufficient concentration of nanoparticles in order to induce photothermal ablation; however, as noted later, internalization of the VMWNPs is not mandatory for inducing sufficient hyperthermia to result in cell death. There were 12.5, 76, 39 and 45% reductions in viable OxS CT-26 cells with 120, 180, 240 and 300 s of laser exposure. Contrary to the OxS CT-26 results, OxR CT-26 cells with internalized VMWNPs exhibited an increase in cell viability both with and without laser exposure. Only at 3 W for 300 s was there a profound reduction in viable cells (62%) (Appendix A).

### 3.8. Photothermal Effect of Extracellular VMWNPs

The photothermal effect of extracellular VMWNPs was evaluated by incubating cells with 0, 25, 50 and 100 μg/mL of VMWNPs and immediately exposing them to 800 nm (1 W) laser irradiation for 60 s or 120 s. With 60 s of laser exposure to the OxS CT-26 cells, there was no significant drop in cell viability up to 50 μg/mL, whereas a considerable reduction (52%) was noticed for 100 μg/mL (Figure 4a). Upon 120 s of laser stimulation, 50 μg/mL and 100 μg/mL produced almost 92% and complete ablation, respectively, whereas no effect was observed with the 25 μg/mL of VMWNPs (Figure 4a). For the OxR CT-26 cells, 60 s of laser stimulation resulted in no significant change in cell viability up to 50 μg/mL, but produced a 54% reduction in cell viability at 100 μg/mL (Figure 4b). Two minutes of laser irradiation produced complete ablation with the 50 and 100 μg/mL of VMWNPs, but there was no difference at 25 µg/mL (Figure 4b). For OxS RKO cells, a 68% decrease in cell viability was observed with 50 μg/mL and complete ablation with 100 μg/mL (Figure 5a), with similar results for 60 or 120 s exposure. OxS RKO cells had an increase in cell viability with 60 or 120 s of 1 W laser stimulation with 0 and 25 µg/mL (Figure 5a). The same trend was observed in OxR RKO cells (Figure 5b). OxR RKO cells treated with 50 µg/mL and immediately exposed to 60 s of a 1 W laser had an increased viability, whereas 120 s led to a 22% reduction. OxR RKO cells treated with 100 µg/mL had 22% and 82% reductions for laser exposures of 60 s and 120 s, respectively. As shown in Figure 5a,b, control cells that were not exposed to the laser or VMWNPs were included because the human cells indicate that laser stimulation alone promotes an increase in cell viability. This trend was also observed in HT-29 cells, which also have no reduction in viability after 60 s of laser stimulation plus VMWNPs for OxS HT-29 cells, even up to a concentration of 100 µg/mL (Appendix A). On the contrary, OxR HT-29 cells had a 33% reduction in viability with 100 µg/mL VMWNPs and 60 s of laser stimulation (Appendix A).

### 3.9. Luminescent Monitoring of Cell Effective Thermal Dose Using VMWNPs

The thermal dose was measured for OxS and OxR CT-26 CRC cells by measuring the loss of luciferase intensity after the photothermal treatment. An incubation with 0, 25, 50 and 100 μg/mL of VMWNPs followed by 60 s or 120 s of 800 nm laser exposure showed an increasing loss of luminescence in both OxS and OxR CT-26 cells (Figure 6a,b). OxS cells exhibited 35% and 75% luminescence loss upon 60 s and 120 s of laser stimulation with 25 μg/mL, respectively, whereas more than a 90% luminescence loss was observed at 50 and 100 μg/mL (Figure 6a). OxR cells showed no loss at 25 μg/mL and 60 s exposure, although a loss of 74% and 95% at 50 and 100 μg/mL was observed (Figure 6b). Two minutes of laser excitation depicted a 36% luminescence loss at 25 µg/mL and a more than 90% loss for 50 and 100 μg/mL.

Cumulative equivalent minutes at 43 °C is a measure of the thermal dose. CEM43 can be calculated using temperature elevations imparted by the VMWNPs. Additionally, CEM43 can be determined by the loss of luminescence following thermal treatments. The calculated value of CEM43 was determined as the sum of CEM43 values calculated at 1 s intervals. CEM43 values for the cells were determined using the luminescence versus CEM43 standard curve. Figure 7a,b demonstrate that the calculated thermal dose closely matches OxS and OxR cells treated with varying concentrations of VMWNPs and either 60 s or 120 s of laser exposure. As observed in cells treated with 100 µg/mL, which induces ablative temperatures, there is a loss of correlation between the calculated and experimental CEM43; however, it is well established that the CEM43 model works well with mild hyperthermia and may fail at ablative temperatures [29,32,39].

### 3.10. VMWNPs-Induced Hyperthermia for Augmenting Chemotherapy

The response of hyperthermia-induced drug uptake was evaluated by exposing OxS and OxR cells to oxaliplatin under the conditions: 37 °C, 42 °C, laser only, NPs only and ‘NPs + laser’ groups. As shown in Figure 8a, these treatments produced no decrease in cell viability in the OxS CT-26 cells in the absence of oxaliplatin. There was a reduction in cell viability with increasing concentrations of oxaliplatin observed for all groups. Exposure to 42 °C for two hours reduced the cell viability compared to the 37 °C, ‘NPs only’ and ‘laser only’ groups with increasing concentrations of oxaliplatin. A greater reduction in cell viability was detected upon laser-induced hyperthermia using VMWNPs. The results of VMWNP plus laser, and 42 °C, at 5 and 25 µM oxaliplatin, resulted in similar reductions (55 and 70%, respectively) (Figure 8a). There was no significant difference between 42 °C and VMWNPs plus laser at 300 µM; however, the nanoparticles provided a statistically significant advantage in the 100 µM group of OxS CT-26 cells. OxS CT-26 cells had decreases in viability for all oxaliplatin concentrations, although this trend was only observed for OxR cells treated with 25, 100 and 300 µM of oxaliplatin. However, there was an advantage to using 42 °C with 5 µM of oxaliplatin in OxR CT-26 cells, resulting in a 49% reduction. The VMWNP photothermal treatment used to induce 42 °C was superior, resulting in an 84% decrease at 5 µM of oxaliplatin. This advantage was not observed at 25 µM (Figure 8b). At 100 and 300 µM concentrations of oxaliplatin, there was a 98% and 99% decrease in viability for the VMWNPs-induced hyperthermia, compared to 87% and 97% for 42 °C. There was a statistical advantage of VMWNP hyperthermia at 100 µM in both OxS and OxR CT-26 cells, but the benefit was lost at 300 µM. It was also observed that the laser alone (minimal heat generation because of the absence of VMWNPs) also resulted in decreases in cell viability compared to 42 °C. Figure 8b shows the combined effect of chemotherapy and hyperthermia on the OxR CT-26 cells. The resistant cells showed no change in viability with the 37 °C, 42 °C, NPs only and ‘NPs + laser’ groups in the absence of oxaliplatin.

The effects of VMWNPs-induced hyperthermia on oxaliplatin efficacy in OxS and OxR RKO cells are provided in Figure 8c,d. Cells were exposed to 0, 5, 25, 100 and 300 μM of oxaliplatin at 37 °C, in the presence of NPs, in the presence of an 800 nm laser (1 W, 60 s), at 42 °C and in NPs + laser treatment groups. In the absence of oxaliplatin, OxS RKO cells showed a 54% drop in cell viability at 42 °C, but no significant effect was detected with the other treatment groups. OxS RKO cells exhibited statistically significant decreases in viability with increasing oxaliplatin concentrations, and the complete obliteration of cells at 100 and 300 µM (Figure 8c). This result is in stark contrast to OxR RKO cells, which only showed sensitivity to 100 and 300 µM of oxaliplatin, regardless of the temperature, presence of NPs or photothermal treatment (Figure 8d). The 42 °C and VMWNPs plus laser groups exhibited almost the same decrease in cell viability (81 and 79%) at 300 µM of oxaliplatin for OxR RKO cells, but the laser stimulation alone also resulted in a substantial reduction.

OxS HT-29 cells begin to be responsive to oxaliplatin at 25 µM. The use of the laser alone, 42 °C for 2 h or 42 °C induced rapidly using VMWNP photothermal treatment provides a greater decrease in cell viability at 100 and 300 µM of oxaliplatin (Figure 8e), although the laser alone once again has benefits and 42 °C was not beneficial at 300 µM. OxR HT-29 cells do not have significant reductions until 300 µM of oxaliplatin. With 300 µM of oxaliplatin, there is a 71% reduction in cell viability with prolonged heating at 42 °C, compared to a 78% reduction with VMWNP-induced 42 °C (Figure 8f). Notably, the human OxR variants are more resistant to oxaliplatin, although hyperthermia at 42 °C can provide benefits at higher concentrations (100 and 300 µM) of oxaliplatin, with 300 µM being a clinically utilized dose [12,17,18,40].

## 4. Discussion

Recently, we have explored the optical properties of different molecular weight fractions of PCPDTBSe and demonstrated the preparation of VMWNPs as a theranostic nanoparticle by combining the NIR emissive oligomer and photothermal high MW fraction. The rapid recombination of electron–hole pairs along the polymer backbone is responsible for heat production from the high MW fraction upon laser stimulation. The aim of combining these two fractions of PCPDTBSe into one nanoparticle was to generate a theranostic material for fluorescence imaging, as well as photothermal therapy, confirmed through the presence of the absorption peaks of the oligomer and high MW PCPDTBSe in the absorbance spectrum of VMWNPs. VMWNPs showed a similar emission spectrum to the oligomer NPs, but with a reduced intensity due to the spectral overlap of the oligomer emission and the high MW absorbance. The VMWNPs manifested photothermal properties, with increasing concentrations sufficient to either ablate cancer cells or induce mild hyperthermia for augmenting chemotherapy. The photothermal conversion efficiency of the VMWNPs was determined to be 46%, which is good for photothermal applications. One of the challenges with theranostic nanoparticles based on polymers that can interface within the nanoparticles is the impact of fluorescence quenching, which drastically reduces the QY [20,41]. VMWNPs could be visualized in the peritoneal cavity of a mouse, which is valuable for determining their localization to CRC micrometastases that have disseminated throughout the abdomen. The standard indocyanine green (ICG) filter in the IVIS system allows for the facile detection of VMWNPs. We have previously demonstrated that intravenously delivered polymer nanoparticles can be visualized and the amount localized to a tumor can be quantified using IVIS [20]. ICG is a common fluorescent dye that has been used in angiography and more recently in nanoparticles for the detection and photothermal therapy of tumors. However, ICG does not have a great photothermal conversion efficiency at 15.4%, and free ICG dye has a quantum yield of 2.7% [42,43]. In addition, ICG and similar fluorescent dyes can photobleach, whereas semiconducting polymers do not [44]. Fluorescence quenching can be overcome by using molecular spacers between the polymer chains so that quenching will be minimized and photothermal efficiency preserved, and this technique has been used to boost ICG nanoparticle fluorescence by up to 16.8% [45]. Although we did not compare the VMWNPs photothermal performance directly to metal nanoparticles, we have directly compared similar polymer nanoparticles to gold and provided a thoughtful comparative analysis in a recent review article [20,44]. ROS generation by VMWNPs appears to stem from the oligomer fraction, and may lead to cytotoxicity. VMWNPs produced no cytotoxicity in CT-26 cells until higher concentrations were reached. On the other hand, both OxS and OxR RKO cells indicated VMWNPs were cytotoxic even at concentrations as low as 100 µg/mL. Similar to CT-26 cells, both OxS and OxR HT-29 cells were insensitive to VMWNPs’ toxicity at lower doses.

OxS and OxR CT-26 cells showed a greater decrease in cell viability at 42 °C than at 37 °C with increasing oxaliplatin concentrations. OxS RKO cells were more susceptible to oxaliplatin than the CT-26 cells. The OxS RKO cells showed a higher cell death at even 100 μM of oxaliplatin and a complete cell death with 300 μM at 37 °C and 42 °C. The resistant RKO cells were very resistant to oxaliplatin, with toxicity observed only at 300 μM, and hyperthermia seemed to augment this effect. HT-29 cells were not as sensitive to the effects of oxaliplatin, and 42 °C augmented therapeutic efficacy. OxS HT-29 cells were insensitive to VMWNPs photothermal ablation at 100 µg/mL; however, OxR cells were sensitive. Contrarily, OxS RKO cells were highly susceptible to VMWNPs-induced photothermal ablation, although their OxR counterpart was only mildly susceptible. The results indicate that OxS RKO cells may be more sensitive to damage from either thermal or chemical insult compared to the OxR variant.

The effective thermal dose due to VMWNPs-induced photothermal therapy was assessed by correlating the luciferase intensity decreases with increasing concentrations of VMWNPs. The calculated CEM43 fits well with the luminescence loss measurement for low thermal doses in both OxS and OxR CT-26 cells. The goal was to correlate the loss of luminescence for the predictive measurement of decreased cell viability following photothermal treatment. OxS CT-26 cells indicated no loss of viability at a low dose; however, there was RLU loss, which might be indicative of an increased thermal damage that the cells are able to overcome following recovery from photothermal treatment. OxS CT-26 cells treated with 50 and 100 µg/mL and 120 s of NIR stimulation had profound reductions in viability and corresponding RLU. The RLU reduction at 25 µg/mL for OxR CT-26 cells does not correlate with the lack of reduced cell viability; however, RLU reductions for the 50 and 100 µg/mL at 120 s of NIR stimulation do. The results of RLU loss are valuable for indicating that sufficient thermally ablative temperatures have been induced, with elevated temperatures leading to 100% RLU loss immediately after the photothermal treatment and obliteration of viable cells. RLU loss at non-ablative hyperthermia correlates well with calculated values of CEM43. This is a valuable result for nanoparticle-induced hyperthermia because direct intracellular temperatures cannot be measured; however, CEM43 measurements can be correlated with the RLU loss and therefore the intracellular temperature.

Intracellular VMWNPs could easily be identified by their signature red fluorescence, which originates from the oligomeric fraction of PCPDTBSe. We also explored whether intracellular nanoparticles could generate mild hyperthermia or ablation. Laser exposure at higher power and longer times was used to ablate cells, but this only resulted in modest reductions in cell viability. Upon 5 min of laser exposure, a significant reduction in cell viability was observed with the internalized nanoparticles, whereas no decrease in cell viability was observed in cells without NPs. This proved that the photothermal properties of the VMWNPs remained functional inside cells. In the future, the tuning of the intracellular nanoparticle concentrations could enable the generation of intracellular hyperthermia for the enhanced uptake of chemotherapy drugs. Laser stimulation without VMWNPs increased cell proliferation of both OxS and OxR CT-26 cells. NIR stimulation of OxS CT-26 cells with internalized VMWNPs resulted in significant cell killing at shorter times of laser application; however, this was not observed in OxR CT-26 cells, most likely due to the lower concentrations of internalized NPs. The photothermal ablation results (Figure 4 and Appendix A) indicate that (1) internalization of the VMWNPs is not needed for effective treatment, (2) OxR cells might be more susceptible to photothermal ablation techniques that utilize extracellular heating, since they do not internalize as high a concentration as OxS cells, and (3) much higher laser powers and longer times are needed in order to induce the same reduction in cell viability for internalized VMWNPs compared to extracellular. This could present a problem, since higher laser powers can induce off-target (non-tumor) thermal injury.

A low concentration (25 μg/mL) of extracellular VMWNPs was found to generate a 5–6 °C increment in temperature upon laser exposure. Our aim was to corroborate whether VMWNPs-induced hyperthermia (which is a rapid technique) was as effective as 42 °C applied for 2 h (clinically utilized parameters) for increasing the effectiveness of oxaliplatin against CRC cells that were sensitive or resistant to oxaliplatin. In the absence of oxaliplatin, OxS CT-26 cells with NPs, laser only, NPs and laser or 42 °C treatments produced minimal reductions in cell viability compared to the control group (37 °C), reflecting no cytotoxic effect of these photothermal treatments. A combination of oxaliplatin with VMWNPs and laser exposure further reduced the cell survival in comparison to cells incubated with oxaliplatin at 42 °C for two hours. This result indicates that rapid hyperthermia using VMWNPs produces an enhanced chemotherapy outcome compared to heating at 37 °C, and is on par with the results of prolonged heating at 42 °C. One of the fundamental questions regarding hyperthermia is the timing of when it should be applied in order to maximize cancer cell death. Although there is a small therapeutic window in which pre-heating may be beneficial for radiation sensitization, chemotherapy sensitization works best when heat and drugs are combined simultaneously [46,47]. There is no advantage of heating cells with mild hyperthermia (below 45 °C) to aid in chemo-sensitization [48]. For this reason, clinical treatments use heat plus chemotherapy against colorectal cancer together, most often in the peritoneal cavity, where the application of heat oxaliplatin can be easily deployed [5,49,50,51,52,53].

Due to the drug resistance of OxR CT-26 cells, the reduction in cell viability was not as low as the sensitive cells after the two hours of exposure to oxaliplatin at 42 °C. The addition of VMWNPs followed by laser treatment exhibited a remarkable drop in viability as compared to the 42 °C treatment. VMWNPs-induced hyperthermia increased the oxaliplatin efficacy irrespective of the oxaliplatin resistance. This hyperthermia effect was achieved by three laser stimulations of a 1 min duration, compared to two hours of hyperthermia at 42 °C. OxS RKO cell lines were found to be very sensitive to oxaliplatin. OxS RKO cells were not only more sensitive to oxaliplatin but also to 42 °C alone; however, OxR RKO cells showed no profound reduction of cell viability with the treatment groups (37 °C, 42 °C, NPs, laser, and NPs + laser) in the absence of oxaliplatin. Higher concentrations of oxaliplatin enhanced reductions in cell viability in the 42 °C and ‘NPs + laser’ group, indicating effectiveness due to hyperthermia. Both OxS and OxR HT-29 cells are more resistant to oxaliplatin compared to CT-26 or RKO cells. They are also not particularly sensitive to 42 °C, provided either by an incubator or VMWNP-induced hyperthermia. However, at higher concentrations of oxaliplatin, VMWNPs conferred an increased reduction in their viability compared to the more prolonged heating at 42 °C. The results demonstrated here provide the first step in demonstrating the safety and efficacy of VMWNPs for the photothermal eradication of CRC. CRC frequently disseminates throughout the peritoneal cavity in small nodules, and although chemotherapy could be suitable for addressing micro-metastasis, the blood–peritoneal perfusion barrier impedes effective chemotherapy. In order to maximize chemotherapy effectiveness, oxaliplatin is often warmed to 42 °C and perfused throughout the abdomen following surgical debulking to remove larger CRC masses, a technique referred to as heated intraperitoneal chemoperfusion (HIPEC). Our goal here was to demonstrate that VMWNPs could confer an advantage to sensitizing oxaliplatin-resistant CRC. VMWNPs can be perfused throughout the peritoneum to target micrometastases and, upon stimulation with near-infrared light, can generate mild hyperthermia to sensitize the micro-tumors to chemotherapy. Future studies will involve the evaluation of VMWNPs in a murine model of peritoneally disseminated CRC using OxR cells compared to OxS cells, which is similar to our previously established model utilizing fluorescent photothermal nanoparticles for the visualization and treatment of peritoneal carcinomatosis [54].

## 5. Conclusions

VMWNPs were utilized for hyperthermia generation to augment chemotherapy in OxS and OxR CRC cells. This treatment technique performed well even with the resistant CRC cell lines. The NIR fluorescence emission of VMWNPs was used to detect their presence inside CRC cells. The ability of intracellular VMWNPs to generate heat for the photothermal ablation of cancer cells was also demonstrated. Increased chemotherapy effectiveness could be imparted by VMWNP-induced hyperthermia (42 °C) with a total of 180 s of laser exposure, compared to 7200 s of bulk heating at 42 °C. In conclusion, theranostic VMWNPs were utilized for NIR fluorescence imaging, the quantification of their intracellular uptake, photothermal ablation using both intracellular and extracellular VMWNPs and augmenting chemotherapy against oxaliplatin-sensitive and -resistant CRC cells.

## Data Availability

The data presented in this study are available in this article and the Appendix A.

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
