# Peer review of "Variable Molecular Weight Polymer Nanoparticles for Detection and Hyperthermia-Induced Chemotherapy of Colorectal Cancer"

_cancers, 2021, doi:10.3390/cancers13174472_

Round 1

Reviewer 1 Report

The work demonstrates the fabrication of polymeric nanoparticles based theranostics for imaging and hyperthermia and discusses the suitability of these particles for rapid hyperthermia against oxaliplatin resistance cancer cells. Overall, the work adds new knowledge to the field and can be published after addressing minor comments.  

1) I have a concern about the stability of the particles in media because the zeta potential is very small (-5 mV). 

2) Authors mentioned increased in 46% hyperthermia efficiency compared to metal nanoparticles, however, hyperthermia efficiency can not be compared due to the difference in laser wavelength. However, it would be good to include a table showing hyperthermia performance comparison with gold and other metal nanoparticles in the literature. 

Author Response

  • I have a concern about the stability of the particles in media because the zeta potential is very small (-5 mV). 

We thank the reviewer for their thoughtful comment. We did not observe instability in media, and we apologize for not including descriptive language about this result in the previous version of the manuscript. We have also previously published on this type of nanoparticles and demonstrated no instability in serum containing media, even with zeta potentials close to zero. Consideration of stability issues have been addressed in a comment in the revised manuscript.

  • Authors mentioned increased in 46% hyperthermia efficiency compared to metal nanoparticles, however, hyperthermia efficiency can not be compared due to the difference in laser wavelength. However, it would be good to include a table showing hyperthermia performance comparison with gold and other metal nanoparticles in the literature. 

Previous literature has compared photothermal efficiency of polymer nanoparticles to metal nanoparticles and we were unaware of wavelength differences being a considerable variable.  We thank the reviewer for bringing it to our attention, and we have revised the language to eliminate this comparison which might be confusing to some readers.  The scope of the current work was not a review article and we remained focused on evaluating the development and utility of the VMWNPs against CRC, and therefore did not provide a table comparing our polymer NPs to metal NPs.  We have previously made this type of comparison in a review article published last year in Advanced Drug Delivery Reviews, vol 163, pp 40-64, and we have referred the readers to this reference for more information.

Reviewer 2 Report

The authors prepared polymer-based NPs that can be used for both photothermal therapy and fluorescence detection, along with some cell experiments to prove this assumption. But the manuscript lacks experimental completeness, and following issues need to be deal with as well.

1. Some format issues: unit issue (e.g., P3, Line74); There should be a blank space between number and unit (e.g., P3, Line112).

2. P6, Line238. Why did authors use 800 nm laser rather than typical laser wavelengths of 808, 1064, etc?

3. P7, Line263. What is the mechanism of higher drug uptake at elevated temperature? How could the authors prove their assumption?

4. Why was 42 ÌŠC used as a comparison to 37 ÌŠC?

5. Fig2. Why was reduced cell viability observed at 0 uM for OxR RKO cells compared to other cell lines?

6. Fig2d, why is cell viability increased by at 5 uM, 25 uM oxaliplatin compared to 0 uM? That is, when the concentration is high, oxaliplatin has toxicity to cells, cell viability decreased. But when the concentration is low, for example 5, 25uM, the cell viability increased vs control (0). Why?

7. Figs4, in oligoNPs+DPBF+laser group, it seems the generation of singlet oxygen stops increasing after 3 min laser. What is the reason? Why was irradiation time not further increased?

8. Figs5, Why were serum starved cells used for cellular uptake? Why is there a big cellular uptake difference between normal cells and serum starved cells?

9. Fig3, looks like nuclei of OX-S cells were not stained well, or CLSM may have been unfocused. Why wasn't another tool used to identify intracellular uptake, such as Flow cytometry?

10. Figs6a, please explain why the cell viability increased after 240s and 300s of laser compared with 180 s laser.

11. Did the authors do any in vivo experiments to evaluate the polymer-based nanoparticles that can both generate heat and be used for fluorescence detection? Cell viability experiments are important, but it's likewise important to move beyond such studies.

12. Did the authors make a comparison between the photothermal properties of polymer-based NPs vs. metal NPs?

Author Response

  1. Some format issues: unit issue (e.g., P3, Line74); There should be a blank space between number and unit (e.g., P3, Line112).

We apologize for the formatting issues and thank the reviewer for bringing them to our attention.  We have carefully examined the manuscript to correct these, and additional formatting oversights.

  1. P6, Line238. Why did authors use 800 nm laser rather than typical laser wavelengths of 808, 1064, etc?

The nanoparticles’ maximum absorption intensity is 765 nm, and we have a laser in the lab that operates at 800 nm.  This laser system has a tolerance of +/- 10 nm, which allows it to be comparable to 808 nm lasers and sufficiently stimulate the VMWNPs.

  1. P7, Line263. What is the mechanism of higher drug uptake at elevated temperature? How could the authors prove their assumption?

The mechanisms of higher chemotherapy uptake at elevated temperatures include increased cell membrane permeability and metabolism, and in vivo, increased blood perfusion, and this has been proven in pre-clinical and clinical literature.  We have previously demonstrated that rapid hyperthermia from photothermal nanoparticles increases cell membrane permeability, as confirmed using inductively coupled plasma mass spectrometry to evaluate platinum content inside cells. We have revised the manuscript to include a description of the mechanisms and reference to the results and methods that can be utilized for confirming that hyperthermia increases drug uptake.

  1. Why was 42 ÌŠC used as a comparison to 37 ÌŠC?

42 °C is routinely employed with delivery of platinum chemotherapy for treatment of CRC in our clinical facilities, as well as many other clinics around the world.  Given the confirmed success of this temperature, and its safety, we chose this temperature for comparison to normothermia.  We apologize that consideration of an optimal hyperthermic temperature was not sufficiently described in the introduction and we have revised the text to provide more specific details.

  1. Why was reduced cell viability observed at 0 uM for OxR RKO cells compared to other cell lines?

The viability of the 0 uM groups was normalized to 1, and although there is less viability compared to some of the other groups, this is not a reduction, but rather the baseline standard. The observed effect is a result of hormesis, which is explained more clearly in response to the next query.

  1. Fig2d, why is cell viability increased by at 5 uM, 25 uM oxaliplatin compared to 0 uM? That is, when the concentration is high, oxaliplatin has toxicity to cells, cell viability decreased. But when the concentration is low, for example 5, 25uM, the cell viability increased vs control (0). Why?

Hormesis is an effect where the insulting agent, in this case chemotherapy with and without hyperthermia, may lead to an increase in cell survival or increased proliferation (stimulatory effects).  This is a common effect observed in toxicology studies, and also in response to some chemotherapy.  The data in Figure 2d indicate that specifically for the OxR RKO cells there is a hormetic effect where oxaliplatin at low doses cannot overcome the drug resistance barrier for toxicity, and in fact induces a stimulatory effect; hence there is an observed trend of increased proliferation.  We evaluated the OxR RKO cells in triplicate in 3 separate experiments because we also thought this result was odd.  All 3 replicates of the experiment showed the same result.  We then proceeded to explore whether the phenomenon was seen in another human CRC line with developed resistance to oxaliplatin, but this trend was only observed in the OxR RKO cell line.  We have expounded upon this result in the text to rationalize the observations and refer the reader to the concept of hormesis. It is an unusual result that may further support the clinically observed trends of disease recurrence due to drug resistance and we have not found such information in previous literature for drug resistant colorectal cancer.

  1. Figs4, in oligoNPs+DPBF+laser group, it seems the generation of singlet oxygen stops increasing after 3 min laser. What is the reason? Why was irradiation time not further increased?

Figure S4 is a preliminary result to try and identify the mechanism of oligomer toxicity.  This result needs to be expounded upon further in future work, but since this mechanism is not the focus of the current work, the result was noted in the supplementary information as a potential mechanism. Since differences between the nanoparticles were noted at 2 cycles of 1 minute of laser exposure there was not an advantage to performing more than 4 cycles of laser heating and cooling to examine the trend.

  1. Figs5, Why were serum starved cells used for cellular uptake? Why is there a big cellular uptake difference between normal cells and serum starved cells?

Serum starving has been used by other groups to promote cell uptake of nanoparticles and other agents, and thus we explored the potential of serum starving to further increase intracellular uptake of the VMWNPs.  We have revised the text to elaborate on why this was done and refer the reader to references, including one published in Cancers, where serum starving improved the uptake of silica nanoparticles.

  1. Fig3, looks like nuclei of OX-S cells were not stained well, or CLSM may have been unfocused. Why wasn't another tool used to identify intracellular uptake, such as Flow cytometry?

The aim of the work was to demonstrate that VMWNPs had been internalized within CRC cells and CLSM provided the most straight forward mechanism of visualization, and hence flow cytometry was not used. We have previously used flow cytometry for detection of intracellular nanoparticles, but in the present case we really wanted to see the location of the intracellular nanoparticles.  The DAPI staining simply shows the location of the nuclei compared to the VMWNPs.  The cells were in focus, although the gain was perhaps set a bit too high for optimal visualization of the nuclei.

  1. Figs6a, please explain why the cell viability increased after 240s and 300s of laser compared with 180 s laser.

There were different cell populations receiving the internalized VMWNPs and laser exposure. As demonstrated in Figure 5, when there are insufficient NPs photothermal ablation may not occur, leading to increased cell proliferation. The results in OxS cells in Figure S6 show increased proliferation in the 240 and 300 s laser stimulation groups, which indicates that perhaps these cells might not have internalized sufficient NPs for ablation.  However, other results in this manuscript show that internalization of the NPs is not needed for improving chemotherapy response.  We have revised the text to better explain this result.

  1. Did the authors do any in vivo experiments to evaluate the polymer-based nanoparticles that can both generate heat and be used for fluorescence detection? Cell viability experiments are important, but it's likewise important to move beyond such studies.

The reviewer raises a very important point, and we have previously published on a similar theranostic polymer nanoparticle for in vivo detection and treatment of CRC. The previous work in mice was not done in OxR cells (only OxS cells were used for tumor delvelopment), and the purpose of the current work was to first demonstrate the difference that VMWNPs have on sensitizing oxaliplatin resistant cells to chemotherapy, using focal hyperthermia. We have revised the manuscript to refer the reader to our in vivo data and expound upon future applications.

  1. Did the authors make a comparison between the photothermal properties of polymer-based NPs vs. metal NPs?

The current article is composed of original data and comparison to metal nanoparticles was not needed to demonstrate the utility of the VMWNPs and differential response of OxS compared to OxR cells. We have previously compared polymeric photothermal NPs to metal NPs in a review article and refer the readers to those works in the revised manuscript.

Round 2

Reviewer 2 Report

The authors have adequately answered most of the questions. Still, there remain several issues.

Given the limited novelty of the paper focus, and the fact that most of the data relate to in vitro cell viability, in vivo data related to hyperthermia and fluorescence imaging using the authors' NPs (VMWNPs) should be included.

Also, the authors note the photothermal efficiency of the NPs was ~46%. However, there is no discussion about the physicochemical tradeoffs between hyperthermia and fluorescence. Given the intrinsic tug-of-war between these, and the focus of this manuscript on the ability of the NPs to produce both heat and flurorescence, this should be carefully discussed in the Discussion.

Also, while the article doesn't read poorly, readability might be improved if the verbosity were reduced.

Author Response

Given the limited novelty of the paper focus, and the fact that most of the data relate to in vitro cell viability, in vivo data related to hyperthermia and fluorescence imaging using the authors' NPs (VMWNPs) should be included.

We respectfully appreciate the reviewer’s point of view and request for in vivo work. However, the scope of the current work was to demonstrate efficacy of the VMWNPs specifically against oxaliplatin resistant CRC compared to oxaliplatin sensitive CRC, which is critical data needed before moving into animal work.  We have begun in vivo work using oxaliplatin resistant CT26 cells, and the model is a bit challenging due to the increased aggressive nature of the resistant cell line.  Given the current focus, the results support the hypothesis that VMWNPs can effectively provide a photothermal mechanism for ablating chemotherapy resistant cells.  We will work diligently to demonstrate the in vivo utility of these nanoparticles, but given the time commitment for such work, need to withhold that data for a future manuscript, as it is a large body of work. We have commented upon the future intent in the revised manuscript.

Also, the authors note the photothermal efficiency of the NPs was ~46%. However, there is no discussion about the physicochemical tradeoffs between hyperthermia and fluorescence. Given the intrinsic tug-of-war between these, and the focus of this manuscript on the ability of the NPs to produce both heat and flurorescence, this should be carefully discussed in the Discussion.

The reviewer has raised a valuable point, and we apologize that we did not describe the relationship between photothermal efficiency and quantum yield for the VMWNPs. We have previously calculated the quantum yield according to published protocols and published that result last year.  We have revised the text of the current manuscript to include the quantum yield and refer the reader to the complete methodology as a reference.  Following this description, we have revised the text to include the challenges associated with managing the structure of the nanoparticle in order to increase the QY while still retaining high photothermal efficiency.  Our team is working on a publication specific to producing an optimized particle.

Also, while the article doesn't read poorly, readability might be improved if the verbosity were reduced.

We thank the reviewer for their thoughtful suggestion and we have revised the manuscript to better streamline the language. 

Round 3

Reviewer 2 Report

The authors have improved the article, but there is still concern about novelty. We understand that a resistant tumor cell in vivo model may be challenging, yet authors should have done other experiments in their stead to improve rigor of this manuscript. For example, these sorts of experiments could suffice:
  1. In part on augmented response to oxaliplatin using hyperthermia. The authors could add 2 control groups to improve: a. the response of cell lines to oxaliplatin without using hyperthemia to assess whether there is a big different between the oxaliplatin response with/without use of hyperthemia; b: How long will the augmented response to oxaliplatin last? Authors could co-culture NPs (without drug) with cell lines, then using hyperthermia on cells, keep co-culturing cells for another ~24 h, then dose cell line with oxaliplatin and test cell viability.

  2. In the part on fluorescence detection. a. The authors could simply use common mice rather than the resistant cell model to assess whether VMWNPs can be detected in vivo after tail vein injection. b. a rough approximation of the circ half life of VMWNPs by testing the fluorescence in blood at a number of time points. c: also make a comparison between VMWNPs and commercial dyes to test whether  VMWNPs is better or comparable.

Author Response

The authors have improved the article, but there is still concern about novelty. We understand that a resistant tumor cell in vivo model may be challenging, yet authors should have done other experiments in their stead to improve rigor of this manuscript. For example, these sorts of experiments could suffice:

  1. In part on augmented response to oxaliplatin using hyperthermia. The authors could add 2 control groups to improve: a. the response of cell lines to oxaliplatin without using hyperthemia to assess whether there is a big different between the oxaliplatin response with/without use of hyperthemia; b: How long will the augmented response to oxaliplatin last? Authors could co-culture NPs (without drug) with cell lines, then using hyperthermia on cells, keep co-culturing cells for another ~24 h, then dose cell line with oxaliplatin and test cell viability.
  1. The response of each of the cell lines, and their respective oxaliplatin phenotypes, to increasing concentrations of oxaliplatin at normothermia (37C), compared to hyperthermia (42C) is provided in Figure 2. There are statistically significant differences in response to hyperthermia, which are especially notable in the developed resistant lines.  This data was done according to clinical utilization of hyperthermia for CRC, which uses 42C for 2 hrs in combination with oxaliplatin. Figure 8 also includes control cells treated at 37C, for comparison to treatment with oxaliplatin at 42C for 2hr or 42C induced using photothermal treatment.

  1. I think what the reviewer is asking for whether the augmented responsiveness to chemotherapy will be evident at some point after the thermal treatment. However, it has already been established that hyperthermia given before chemotherapy is not effective, and this result has specifically been shown for CRC cells. Previous research has lead to clinical utilization of hyperthermia that is always simultaneously given with chemotherapy in the clinical setting in order to impart a beneficial therapeutic response.   Pre-treatment with hyperthermia only works well for radiation sensitization. We have revised the text to describe these effects and to include supporting references.  Since hyperthermia (42C) pre-treatment has already been shown by other authors not to enhance chemosensitization of oxaliplatin in chemo-sensitive lines, pre-treatment is unlikely to yield benefit in chemo-resistant lines.  Prolonged hyperthermia is toxic to all cells, therefore localized mild hyperthermia delivered for a few hours is used clinically, most often in regional perfusion modalities like limb and peritoneal perfusion.  Specifically for oxaliplatin, heat and oxaliplatin are provided simultaneously during heated intraperitoneal chemoperfusion (HIPEC), which is a widely used clinical technique.  The goal of using nanoparticles is to sensitize cells to chemotherapy with rapid delivery of mild hyperthermia compared to the routine use of hours of elevated temperature.  The nanoparticles only generate hyperthermia with exposure to light and they are inert otherwise.  Using the nanoparticles to rapidly generate hyperthermia and then culturing the cells with nanoparticles for 24hrs before exposure to chemotherapy will therefore not offer any benefit.    
  1. In the part on fluorescence detection. a. The authors could simply use common mice rather than the resistant cell model to assess whether VMWNPs can be detected in vivo after tail vein injection. b. a rough approximation of the circ half life of VMWNPs by testing the fluorescence in blood at a number of time points. c: also make a comparison between VMWNPs and commercial dyes to test whether  VMWNPs is better or comparable.
  1. We have previously published that a very similar semiconducting polymer nanoparticle is detectable in solid breast tumors 24hrs after injection. We have included this reference and revised the text to explain and guide the reader.  This work also quantified the amount of nanoparticles that localize to the tumor, which addresses part b of the reviewer’s query. Development of a single tumor nodule and evaluation of VMWNPs localization is a bit removed from the clinical case, where CRC develops in many small micrometastatic lesions. We are specifically working on peritoneally disseminated lesions, and we have revised the text to indicate that the developed nanoparticles will be delivered via peritoneal perfusion, as our goal is to mimic the HIPEC technique.  We have tried diligently to address the reviewer’s query of whether the VMWNPs can be visualized at depth in an animal model.  We procured a freshly euthanized Balb/C mouse without tumors (had been a female breeding mouse) and injected 100 uL of 250 ug/ mL of VMWNPs interperitoneally and imaged the animal by IVIS to detect the fluorescence.  We have included this result in a new figure in the supplementary information to demonstrate that the VMWNPs can be observed easily in vivo. 
  2. We appreciate the reviewer’s consideration for requesting circulation data. However, evaluating the circulating half-life of the VMWNPs in blood might be more suitable for other tumor types. We are focused on colorectal cancer, which often has chemotherapy resistant metastases disseminated throughout the peritoneal cavity.  Due to the blood-peritoneal perfusion barrier chemotherapy is not as effective as in other metastatic cancer scenarios.  Therefore techniques like HIPEC following surgical debulking are often used. We have revised the text to make it more clear how we envision the VMWNPs to be used for peritoneally disseminated CRC.  Our group is currently working on an animal model with functionalized nanoparticles based on the VMWNPs platform, and we disclose the results for the reviewer only, as they are premature for publication. The animal model uses the developed OxR CT26 cells, but they develop into tumors very rapidly in vivo and we are learning how many to provide to develop well-dispersed disease. We used a perfusion circuit to provide targeted (AcTZ) versus untargeted (OH) nanoparticles to evaluate co-localization of nanoparticles to CRC.  We then used a coliseum technique and a laser light scattering agent to expose the entirety of the abdomen to laser light to stimulate hyperthermia in the absence of chemotherapy. As the reviewer can see we still need to optimize targeting and we are working on this aspect.  We have quantified the amount of nanoparticles that exit the perfusion circuit and are hence are unbound.  We provide this content for the reviewer so they may appreciate how the results in the manuscript are being extended and to better explain the vision since these particles will not be delivered via tail vein, but they can be quantified via fluorescence in the peritoneal perfusion modality.
  3. We have revised the text to compare how VMWNPs can be compared to fluorescent dyes, including indocyanine green (ICG), which can be used for detection, but not for photothermal therapy in their current free-floating form. This includes discussion of one of ICG nanoparticles which can be used for photothermal and also explains the problems with photobleaching that occurs with dyes and which semi-conducting particles can overcome.
